# Antibacterial Capability of MXene (Ti_3_C_2_T_x_) to Produce PLA Active Contact Surfaces for Food Packaging Applications

**DOI:** 10.3390/membranes12111146

**Published:** 2022-11-15

**Authors:** Xiomara Santos, Marcos Álvarez, Diogo Videira-Quintela, Aranzazu Mediero, Juana Rodríguez, Francisco Guillén, Javier Pozuelo, Olga Martín

**Affiliations:** 1Escuela Politécnica Superior, Departamento de Ciencia e Ingeniería de Materiales e Ingeniería Química, Universidad Carlos III de Madrid, Avenida Universidad 30, 28911 Leganes, Spain; 2Departamento de Química Analítica, Química Física e Ingeniería Química, Facultad de Farmacia, Universidad de Alcalá, Ctra. Madrid-Barcelona km 33.6, 28871 Alcala de Henares, Spain; 3Instituto de Investigación Sanitara Fundación Jimenez Diaz, Avd. Reyes Católicos 2, 28040 Madrid, Spain; 4Departamento de Biomedicina y Biotecnología, Facultad de Farmacia, Universidad de Alcalá, Ctra. Madrid-Barcelona km 33.6, 28871 Alcala de Henares, Spain

**Keywords:** PLA, MXene, bactericidal activity, food packaging, antibacterial surface, food contact material

## Abstract

The globalization of the market and the increase of the global population that requires a higher demand of food products superimposes a big challenge to ensure food safety. In this sense, a common strategy to extend the shelf life and save life of food products is by avoiding bacterial contamination. For this, the development of antibacterial contact surfaces is an urgent need to fulfil the above-mentioned strategy. In this work, the role of MXene (Ti_3_C_2_T_x_) in providing antibacterial contact surfaces was studied through the creation of composite films from polylactic acid (PLA), as the chosen polymeric matrix. The developed PLA/MXene films maintained the thermal and mechanical properties of PLA and also presented the attractive antibacterial properties of MXene. The composites’ behaviour against two representative foodborne bacteria was studied: *Listeria mono-cytogenes* and *Salmonella enterica* (representing *Gram-positive* and *Gram-negative bacteria*, respectively). The composites prevented bacterial growth, and in the case of *Listeria* only 0.5 wt.% of MXene was necessary to reach 99.9999% bactericidal activity (six log reductions), while against *Salmonella*, 5 wt.% was necessary to achieve 99.999% bactericidal activity (five log reductions). Cy-totoxicity tests with fHDF/TER166 cell line showed that none of the obtained materials were cytotoxic. These results make MXene particles promising candidates for their use as additives into a polymeric matrix, useful to fabricate antibacterial contact surfaces that could prove useful for the food packaging industry.

## 1. Introduction

The food safety along the supply chain is still an ongoing challenge for developed countries. The lack of proper safety control methods to tackle microbial invasion due to contamination often leads to the occurrence of food-borne diseases that pose a serious threat to public health, due to their association with an increase in morbidity and mortality, as well as to the country’s economy. Microorganisms such as *Listeria* can survive and grow at refrigeration temperatures, both in aerobic and anaerobic conditions, while *Salmonella* is resistant to temperatures between 5 and 12 °C [1]. *Listeria* and *Salmonella* are among the most important foodborne pathogens with a major economic impact on global public health [2,3,4].

Among the foodborne illnesses, listeriosis is the one with the highest lethality rate, and more than 60% of people who overcome this disease have neurological sequelae [2,5,6]. A wide variety of food may contain *Listeria* such as soft cheese, meats, unpasteurized milk, smoked seafood, etc. [2]. On the other hand, salmonellosis, sometimes underestimated because it can be asymptomatic [7], has become the second most frequently reported zoonotic disease in humans [3]. *Salmonella enterica* is a genus of bacteria frequently present in the intestines of healthy birds, reptiles, and mammals [7]. In 2019, 17.9% of the reported foodborne cases in the European Union were salmonellosis, which means a total of 87,923 infected people [3].

There are several approaches to obtain food contact materials (FCM) with antibacterial properties that prevent contamination of the FCM itself and more importantly of the food surface. For this reason, the control of the above-mentioned pathogens is of special interest to ensure food safety. In this sense, the use of active food contact materials is gradually taking importance within the packaging industry. This type of materials is developed in such a way that it interacts with the food or its environment, with the goal of preventing oxidation, microbial attacks, or humidity problems [1,8]. 

As antibacterial inorganic substances, MXene (Ti_3_C_2_T_x_) were selected, which are a relatively new family of two-dimensional inorganic materials made up of carbides or nitrides obtained from the selective exfoliation of the “A” layers of the MAX phases. These materials have a high specific surface area, and good electrical conductivity, hydrophilicity, and antibacterial activity, which has promoted their use in a wide range of applications, both from the electronic and sensor point of view, as well as for biomedical applications [9,10,11]. Therefore, they have become one of the most versatile materials during the last decade, whose synthesis, properties and the exploration of new applications have become one of the areas of greatest interest in science and technology [10].

In the literature, we can find works that prove the antibacterial activity of MXenes against both Gram-positive and Gram-negative bacteria. Different mechanisms have been proposed to explain this attractive property. In the first place, it has been found that the direct contact of the sharp edges of MXene nanosheets can cause the deformation and/or rupture of the cell membrane of the bacteria [12,13]. On the other hand, bactericidal activity has been related to the oxidative stress caused by the generation of reactive oxygen species (ROS) [13]. Additionally, due to the good conductivity of MXenes, another mechanism of action could take place due to the formation of a conductive bridge that allows the transfer of electrons between the interior and exterior of the cell, producing cell death [12]. In another study, Liu et al. obtained highly sensitive multifunctional electronic skin based on nanocellulose/MXene composites. In addition, to demonstrate the biocidal properties of these materials against *Escherichia coli* and *Staphylococcus aureus*, they determined the cytotoxicity against the L929 cell line, showing adequate biocompatibility [14].

There is a global goal to replace traditional polymers (e.g., polypropylene, ethylene vinyl alcohol, or polyethylene) that are non-biodegradable and come from non-renewable sources such as petroleum, generating a negative impact on the ecosystems [15,16]. Nowadays, a focus on the use of sustainable materials, obtained from natural, renewable, biodegradable and non-toxic sources, is a common strategy. This new path represents a positive impact from both the economic and ecological point of view [16,17], and several FCMs derived from polylactic acid (PLA) [18], sugar cane [19,20], starch [18,21], pectin [22,23], chitosan [24,25] and nanocellulose [26,27] have been studied to obtain sustainable packaging. 

PLA constitutes a polyester that has acquired vital importance for biomedical applications and in the food packaging industry due to its biodegradability, biocompatibility, excellent mechanical properties and its ability to be industrially compostable [16,28]. Due to all the characteristics mentioned so far of PLA, today it is considered a safe material for applications in the food packaging industry [2].

Although the biodegradability and compostability are sometimes considered synonymous, it is important to understand that if the conditions for biodegradability are not optimal, it may not occur in a reasonable period of time and it will be necessary to accelerate the biodegradation process under certain conditions. PLA is a polymer whose biodegradation occurs under specific conditions, resulting in degradation products that do not constitute environmental contaminants. Usually, during the degradation of PLA, ester bonds are first broken to produce oligomers, dimers, and monomers that, due to their small size, are able to cross the cell walls of microorganisms and, therefore, be degraded by microbial enzymes [29,30,31]. For this reason, PLA waste is industrially treated under composting conditions, with the objective of accelerating the biodegradation process of the polymer. 

The incorporation of additives into the PLA is known to affect the biodegradability process, for example, in a PLA/TiO_2_ system, in which the authors reached the conclusion that the addition of the particles to the polymeric matrix causes an increase in the rate of degradation of the PLA [31].

Based on the mentioned properties with respect to MXene-type Ti_3_C_2_T_x_, observed in the mentioned works in the preceding paragraphs that prove adequate antibacterial activity and biocompatibility, these structures were selected as promising candidates to be used as inorganic fillers to obtain antibacterial contact surfaces with potential applications as food contact materials. PLA was the chosen polymeric matrix in order to evaluate the active role of the MXene-type Ti_3_C_2_T_x_, also having in mind the sustainability of the chosen polymer matrix. Despite the fact that the addition of inorganic fillers such as TiO_2_, ZnO or Ag into a PLA matrix is a common practice to develop antibacterial systems [32,33,34,35,36,37,38,39,40], we are not aware of investigations comprising PLA/MXene systems for the same purpose.

The present work evaluates the suitability and active role of MXene (Ti_3_C_2_T_x_) as alternative inorganic fillers, to develop antibacterial systems useful to prevent bacterial contamination. For this, a system comprised of PLA with different amounts of Ti_3_C_2_T_x_ nanofiller was synthesized and characterized in terms of its mechanical and thermal properties to evaluate the influence of the MXene on the PLA matrix. To assess the antibacterial role of the MXene incorporated into the PLA matrix, the antibacterial properties of the developed composites was studied against two representative foodborne microorganisms: *L. monocytogenes* and *S. enterica*. In addition, the cytotoxicity of the developed composites was evaluated to assess the possible toxicity level of using MXene-based composites in a human-contact application, such as the food contact material.

## 2. Materials and Methods

### 2.1. Materials

MAX phase (Ti_3_AlC_2_) (Carbon-Ukraine Co., Kiev, Ukraine), lithium fluoride (Sigma-Aldrich, Co., Burlington, MA, USA), and hydrochloric acid (Sigma-Aldrich Co., Darmstadt, Hesse, Germany) were used in the synthesis of the filler particles, MXene. In addition, to obtain the composites, a poly lactic acid matrix, (PLA, IngeoTM Biopolymer 2003D NatureWorks, Plymouth, MN, USA) and dichloromethane (labkem, Barcelona, Catalonia, Spain) were used. *S. enterica* (ATCC 35664) obtained from the American Type Culture Collection (ATCC, Washington D.C., MD, USA) and *L. monocytogenes* obtained from a sample retrieved in the University Hospital Príncipe de Asturias were used in the antibacterial activity tests. fHDF/TER166 human fibroblasts (Evercyte, Vienna, Austria) were used in the cytotoxicity assay.

### 2.2. Methodology for MXene (Ti_3_C_2_T_x_) Synthesis

For the MXene (Ti_3_C_2_T_x_) synthesis, Ti_3_AlC_2_ MAX phase was used as starting raw material. The aluminum layer was removed with hydrofluoric acid obtained in situ by adding 2 g of lithium fluoride to 20 mL of 9 mol/L hydrochloric acid solution, to which 1 g of MAX phase was then slowly added. The reaction was kept under constant magnetic stirring for 48 h at 35 °C. Once the reaction was completed, several washing cycles were carried out in a centrifuge (Rotofix 32A; Hettich zentrifugen; Tuttlingen, Baden-Wurtemberg, Germany) at 6000 rpm for 45 min until the pH of the medium was close to 7. To achieve adequate separation of the MXene layers, the resulting dispersion after washing was kept under stirring for 48 h in a vortex equipment. The supernatant was separated in a centrifuge for 5 min at 4000 rpm, frozen and finally lyophilized to obtain the Ti_3_C_2_T_x_ MXene filler.

### 2.3. Methodology for the Preparation of PLA/MXene Composite Films

PLA/MXene composite films were prepared at room temperature by a solution-casting method using dichloromethane as solvent. To do this, dispersions in dichloromethane of the Ti_3_C_2_T_x_ particles (0.1, 0.3, 0.5, 1, 3, and 5 wt.% with respect to the polymer mass) were obtained and then the PLA was added up to 10% of mass with respect to the solvent. The adequate dispersion of the MXene nanosheets in the polymeric matrix was achieved by applying an ultrasonic treatment. To avoid the heating of the sample or its possible degradation by this process, it was carried out at 10 intervals of 10 s of ultrasonic radiation followed by 50 s without radiation. To achieve homogeneity in film thickness, a 1000 µm-thick Doctor Blade was used on a glass support. PLA composite films were obtained with approximately a thickness of 50 µm. The composite films were dried at room temperature for 24 h and then placed in an oven at 40 °C for 5 days to achieve a complete removal of the solvent [41].

### 2.4. Material Characterization

#### 2.4.1. MXene Particles

**Structural characterization** of synthesized filler MXene was carried out by a field emission scanning electron microscope (FESEM; Hitachi SU-70; Chiyoda, Tokyo, Japan) for morphology and XRD for crystallinity. The acceleration voltage in the analysis of the lyophilized MXene was 10 kV. X-ray diffraction was done by a Philips X’Pert-MPD equipment (Amsterdam, North Holland, Netherlands). The 2θ scan range was set between 5° and 45° with a step size of 0.035°. Additionally, the current intensity and voltage were set at 40 mA and 40 kV, respectively. To perform the test, the samples were dispersed in acetone.

**Antibacterial properties** of MXene against *L. monocytogenes* (Gram-positive) and *S. enterica* (Gram-negative) were carried out by determining by the minimum bactericidal concentration (MBC), following a protocol based on the ISO 20776-1:2006 [42]. Briefly, dispersions in water with concentrations of 13, 8, 4, 2 and 1 mg/mL of MXene were prepared by an ultrasonication process. The bacteria were grown on plate count agar (PCA) plates for 24 h at 37 °C. Some colonies were then collected and suspended in 20 mL of Muller-Hinton broth medium and shaken for 24 h at 37 °C. The following day, the concentration was adjusted to 10^8^ colony-forming units, (cfu)/mL (absorbance between 0.08–0.11 at λ = 625 nm) followed by dilution to 2 × 10^7^ cfu/mL. The assay was performed in multiwell plates. During 24 h of agitation at 37 °C, 100 µL of the MXene dispersion, 100 µL of the double-concentrated Muller-Hinton medium and 5 µL of the bacteria interacted. After 24 h, 10 µL of the resulting dispersions was inoculated onto PCA plates and grown for 24 h at 37 °C. Finally, the bacteria growth was evaluated on the PCA plates.

#### 2.4.2. PLA/MXene Composites

**Dispersion characterization** of filler in PLA matrix and **qualitative composition analysis** was made by FESEM (Hitachi SU-70; Chiyoda, Tokyo, Japan) with energy dispersive spectroscopy (EDS).

**Thermal characterization** was carried out by dynamic scanning calorimetry (DSC) and thermogravimetric analysis (TGA). DSC thermograms were made using METTLER TOLEDO DSC equipment (Greifensee, Zürich, Switzerland). The thermal treatment was performed from 25 to 180 °C. The heating rate was 10 °C/min and a nitrogen atmosphere was used. Two heating–cooling cycles were performed to eliminate the thermal history of the material. TGA was carried out in a Perkin Elmer model STA 6000 equipment (Waltham, MA, USA) through a heating cycle from 50 to 600 °C at a rate of 30 °C/min. The assay was performed in an aluminum boat under nitrogen flow of 20 mL/min.

**Mechanical characterization** was obtained by tensile tests to get the typical stress vs. strain curves. These assays were performed on a TA Instrument dynamo-mechanical analyzer (DMTA), model Q800 (New Castle, DE, USA). Films of 20 × 2 mm (length, width) were fixed with striped clamps and a force ramp of 3 N/min up to 18 N was applied to them to cause the fracture of the specimen. The study was carried out under isothermal mode at 35 °C.

**Cytotoxicity** was evaluated using fHDF cells, seeded at a concentration of 10,000 cells/cm^2^ on 48-well plates with Dulbeccos’ modified essential medium (DMEM, Invitrogen, Thermo Fisher Scientific Inc., Waltham, MA, USA) containing 10% fetal bovine serum and 1% penicillin–streptomycin. Cells were incubated overnight at 37 °C and 5% CO_2,_ then the medium was replaced with fresh DMEM and PLA and composite films (0.3, 1, 5 wt.%) were added. Cytotoxicity was measured 48 h after osteoblastic differentiation promotion using the CytoTox 96^®^ Non-Radioactive Cytotoxicity assay (Promega, Madison, WI, USA) following manufacturer’s protocols (n = 8). 

**Antibacterial activity** of PLA/MXene composites was tested against *L. monocytogenes* and *S. enterica*. This assay was performed by the method described in JIS Z 2801 with some modifications as reported in previous work [43]. Briefly, both bacteria were cultivated in PLA plates for 24 h at 37 °C and after that, some colonies were collected to make a bacterial dispersion in sterile water adjusted to 150 × 10^6^ cfu/mL by UV-Vis at λ = 555 nm (absorbance = 0.125). Next, 100 µL of this dispersion was placed in a sterile PP Petri dish, to which the films (2 × 2 cm) were placed on top and maintained for 24 h at 28 °C. After that, 9.9 mL of sterile water was added, followed by agitation for 20 min. Ten-fold dilutions of the as-prepared dispersion were made and grown on PCA plates for 24 h at 37 °C. The number of colonies was then counted and the ufc/mL was calculated. Three replicates of the blank (PLA) and five of the sample films were made. The results are described as an average logarithm reduction with the corresponding standard deviation by comparison with the bacteria control. The logarithm reduction was determined by calculating the subtraction of the logarithm base 10 of the cfu/mL for the bacteria control, and the logarithm base 10 of the cfu/mL corresponding to the PLA and the composite films.

## 3. Results and Discussion

### 3.1. MXene Filler

Figure 1a shows the appearance of the dispersion supernatant obtained after 48 h of stirring to achieve the separation of the MXene layers. MXene dispersions in water were stable for several days, which is proof of the correct delamination and the hydrophilic surface of the MXene. Freeze-drying of the dispersions was used for two reasons. This drying process avoids the packing of the MXene nanosheets together, which would hinder future dispersion processes, and avoids the oxidation of MXene in aqueous dispersions.

For the FESEM images, 0.3 mL of the dispersion was taken and lyophilized, to obtain a Ti_3_C_2_T_x_ sponge. In Figure 1b we can see the piece obtained after the lyophilization process. In the FESEM image obtained from a cross-section of the sample, the typical porous structure of a sponge was observed (Figure 1c,d). This porous structure is due to the lyophilization process, which was used in the drying of MXene. It is possible to observe how the crystalline growth of water is reflected in the longitudinal and transversal direction of the pores due to the cooling process and because the thin walls are made up of MXene, which shows that the dispersion process has achieved the delamination on the product of the synthesis. Figure 1d shows that the characteristic laminar structure (“like bookbinding or accordion”) of MXene has been broken due to the sought delamination process carried out on the product of the synthesis [44]. Almost free MXene foils were obtained, which were easier to be dispersed in a polymer matrix.

Figure 1e compares the XRD patterns of MAX phase and the synthesized MXene. The characteristic peaks of the lamellar packaging of Ti_3_AlC_2_ can be observed for the crystallographic planes (002), (004) and a small signal for the plane (006), and how they disappear due to the elimination of the lamellar structure in the synthesis of the MXene (Ti_3_C_2_T_x_). A slight signal remains from the peak (002) in the synthesized MXene, although there is a displacement from 9.5° (Ti_3_AlC_2_) to 6.77° (Ti_3_C_2_T_x_) [45,46]. This shift is due to the increase in the interplanar distance caused by the insertion of the hydroxyl groups from the synthesis process and the adhesion of water molecules. It can also be observed how the most intense peak of Ti_3_AlC_2_, the one corresponding to the (014) plane, significantly decreases its value in the diffractogram of Ti_3_C_2_T_x_; this is mainly due to the synthesis process. However, we can determine that some traces of Ti_3_AlC_2_ could remain after the synthesis. These results allow us to confirm that the process of elimination of the aluminum layer and the mechanical delamination used in the synthesis has generated MXene in a satisfactory way.

#### 3.1.1. Antibacterial Activity

Before the addition of the MXene into the PLA matrix, an assessment was first made regarding the antibacterial activity of the filler. The determination of the MBC for the synthesized MXene was necessary since previous studies demonstrated the strong dependence of the concentration of Ti_3_C_2_T_x_ on antibacterial activity [12]. A representative replica of the MBC assay for the MXene particles is shown in Figure 2. The obtained results showed MBC values of 8 and 13 mg/mL of Ti_3_C_2_T_x_ against *L. monocytogenes* and *S. enterica*, respectively.

These results are consistent with the literature that observed higher activity against Gram-positive bacteria [12,13]. Rasol et al. explain that this difference in antibacterial activity is the product of the characteristics of the cell wall of Gram-negative and Gram-positive bacteria. Gram-negative bacteria have an outer protective lipid membrane in the cell wall that could prevent direct damage caused by MXene particles to the plasmatic membrane of this type of bacteria. On the other hand, although Gram-positive bacteria have a thick peptidoglycan layer in the cell wall, the lack of the outer membrane makes them a susceptible target for direct contact with MXenes [12].

Another factor that could explain the differences in the antibacterial activity of MXene against Gram-positive and Gram-negative bacteria is the charge that cell surfaces present. Gram-negative bacteria usually have lower isoelectric points than Gram-positive bacteria. Although the MXene nanosheets are also negatively charged, the more negatively charged surface of Gram-negative bacteria may contribute to the higher repellence of *S. enterica* to the Ti_3_C_2_T_x_ nanosheets [12,13].

### 3.2. PLA/MXene Composites

Figure 3 shows the FESEM images of PLA with 0.3, 1 and 5 wt.% of MXene. It can be observed that in the dispersion process of Ti_3_C_2_T_x_ in PLA, using dichloromethane as the solvent, a high number of agglomerates remains. This is mainly due to the low stability of the MXene dispersions in dichloromethane, which makes the disaggregation of the MXene sheets difficult. However, no larger agglomerates were observed that could impair the mechanical properties of the films as it will be checked. It can also be observed that the size of the agglomerates increased when increasing the amount of MXene, which corroborates the poor interaction between Ti_3_C_2_T_x_ and dichloromethane. 

An elemental surface analysis (EDS analysis) at different white areas (MXene particles) also confirmed the presence of the MXene. For example, the sample with 5 wt.% of Ti_3_C_2_T_x_ had 65%, 23% and 10% of carbon, oxygen and titanium respectively. There are also traces (less than 1%) of other elements such as aluminum, chlorine and fluorine that are not representative, and can be considered as impurities.

#### 3.2.1. Thermal Characterization

**Differential scanning calorimetry** analysis of the samples was performed, and similar behaviour was observed for each of them (Figure 4). In all cases, the T_g_ (glass transition temperature) was above 58 °C with an endothermic peak related to a relaxation process. In addition, at 105 °C, an exothermic process began, corresponding to cold crystallization (T_c_). Finally, at around 142 °C, the melting (T_m_) of the polymer began, representing an endothermic process. 

Table 1 shows the results obtained for the calorimetric analysis of the different samples. As the load of MXene in the polymer increased, an increase in the enthalpy of crystallization was perceived. MXene seems to cause an increase in the crystallization of PLA. This would also explain the increase in the enthalpy of melting of the composites, since by increasing the MXene load, there is a greater amount of crystalline part in the polymer.

**Thermogravimetric analysis** of PLA samples and composites loaded with 0.3, 1 and 5% of MXene helped to determine how the addition of Ti_3_C_2_T_x_ nanoparticles affects PLA decomposition temperature. Table 2 summarizes the decomposition temperatures of each system.

According to the results, it can be concluded that the addition of MXene to PLA produces a slight decrease in the polymer decomposition temperature, and hence its thermal stability. This effect has been observed in other reported systems. Gong et al., attributed the decrease in the decomposition temperature to the local heating that the nanoparticles can generate since they present a high specific surface and chemical activity due to their small size [47]. Nonetheless, highly thermally stable composites were obtained for the proposal application, with maximum decomposition temperatures above 350 °C. 

#### 3.2.2. Mechanical Characterization

The mechanical behaviour of the PLA and the composite films with 0.3, 1 and 5 wt.% of MXene particles is represented by the stress vs. strain curves in Figure 5, and Table 3 shows the elongation at break and tensile strength values for each system. 

According to the results, an increase in MXene concentration in the polymeric matrix causes an increase in the maximum stress and a decrease in strain (Table 3). This phenomenon is translated into an increase in the brittleness of the obtained materials, which was clearly observed in the stress vs. strain curves by the low elongation at break values obtained for the composite films (Figure 5). 

Our results are in agreement with those obtained by Jonoobi et al. [48], who studied the mechanical properties of cellulose nanofiber-reinforced PLA, prepared by twin-screw extrusion, in which it was observed that as the reinforcement increased, the maximum stress and the modulus increased and the maximum deformation experienced by the composites decreased. The high standard deviations of the studied mechanical parameters relate it to the non-homogeneous distribution of the nanofiller in the PLA.

The decrease in the maximum deformation may be due to the lack of interaction between the PLA and the filler (MXene). Peterson et al., explained that this behaviour is due to the addition of rigid reinforcements to the polymer matrix causing stress concentrations. Another possibility could be explained by the weak interaction between PLA and the filler, which could make MXene act as voids inside the polymeric matrix [49].

#### 3.2.3. Cytotoxicity

Cytotoxicity is related to the degree that a compound or chemical can be toxic to cells. This could be due to the rupture of the cell membrane, decreasing cell viability and proliferation. In the case of MXene, the possible release of ROS and the direct interaction of MXene with the cell membrane are explained as reasons for their cytotoxicity [9]. 

Previous works agree that the cytotoxicity of MXene will depend on the synthesis method, the oxidation state, the interaction mechanism, the functional groups on the surface, the size, the route of administration and the exposure time, as well as the cell lines used in the cytotoxicity assay [9]. In the literature, we can find a large number of works based on biomedical applications of MXene. However, there are few in which cytotoxicity and biocompatibility have been studied [9].

In this work, the cytotoxicity of the MXene incorporated into a PLA matrix against the fHDF/TER166 cell line was studied. All data obtained in the cytotoxicity assay were normalized with respect to the control cells. The normality of the data was evaluated using a Lilliefors test and it was found that the data did not follow a normal distribution. Therefore, the median and interquartile ranges were used to describe the results obtained, as shown in Figure 6.

Multiple comparisons were made between all the conditions using a Dunn’s test with a Benjamini-Hochberg correction, concluding that there are no statistically significant differences between the control and the PLA, PLA + 0.3% MXene and PLA + 1% MXene. For the sample with 5% MXene, around 10% cytotoxicity was observed compared to the control. This test could also indicate the low level of migration of MXene in all composites, but, as the loading increases, higher migration may be expected and thus a higher cytotoxicity, as observed in the 5 wt.% composite films. However, as cell viability is above 70%, we can say that none of the tested samples is cytotoxic [50]. 

#### 3.2.4. Antibacterial Activity

The antibacterial results of the PLA/MXene composite films are represented in Figure 7, and a correlation of the antibacterial activity with the loading concentration of MXene particles is observed. Starting from an initial bacterial concentration of a Log (cfu/mL) of 8 (i.e., 150 × 10^6^ cfu/mL) for *L. monocytogenes* and at 0.3% load, antibacterial activity was initially observed. However, it was not until concentrations were of at least 0.5 wt.% that the bactericidal activity could be effectively observed, reaching a log reduction of six that corresponds to 99.9999% of bacterial reduction.

In the case of *S. enterica*, a clear resistance was observed, since only at concentrations of 5 wt.% was a clear bactericidal efficiency observed, with a log reduction of ~5.2 that corresponds to at least 99.999% bacterial reduction. Nonetheless at a 3 wt.% load of MXene a slight antibacterial effect started to appear. These results agree with those obtained from the MBC of MXene against both bacteria, where a higher concentration was needed to kill *Salmonella* (13 mg/mL) than for *Listeria* (8 mg/mL). The possible reasons explained in Section 3.1.1 can also be applied in the case of the results obtained from the composite films for the different bacteria.

The novelty and effectiveness of MXene as antibacterial fillers was compared to other systems listed in Table 4 that are described in function of the logarithm reduction that each system reached for a specific model microorganism. It can be observed that the PLA/MXene system is one of the most efficient and/or highly similar in comparison to common antibacterial inorganic fillers (e.g., Ag, ZnO or TiO_2_). Furthermore, a higher log reduction was also achieved in comparison to other systems that also used MXene as the filler (Table 3). 

The data here presented clearly indicate the active antibacterial role of the MXene when incorporated inside a polymeric matrix, in this case a PLA matrix. Considering that the performed test involved a direct contact between the composite and the selected microorganism, this clearly indicates the suitability of the composite to avoid possible contamination of the surface of a food contact material, thus ensuring food safety as the expected goal.

## 4. Conclusions

MXene (Ti_3_C_2_T_x_) was used as an antibacterial inorganic filler incorporated into a PLA matrix. The resultant PLA/MXene composites showed an increase in the crystallinity of the polymer and a slight decrease in the PLA decomposition temperature. However, in the thermal characterization there was no sign of the possible deterioration of MXene properties, which could favour the insertion of this filler in the packaging industry, where high temperatures are used to process polymers. The mechanical analysis showed an increase in the brittleness of the material. The composites showed biocidal activity against *Listeria monocytogenes* and *Salmonella enterica*, with higher activity against Gram-positive bacteria. The cytotoxicity test showed that the obtained composites are not cytotoxic. This study successfully implemented MXene (Ti_3_C_2_T_x_) as an alternative inorganic filler to develop an antibacterial contact surface. Based on the results derived from this study, we believe that Ti_3_C_2_T_x_ could be used as an additive to obtain active food contact materials (with antibacterial properties) with potential applications in the food packaging industry.

## Figures and Tables

**Figure 1 membranes-12-01146-f001:**
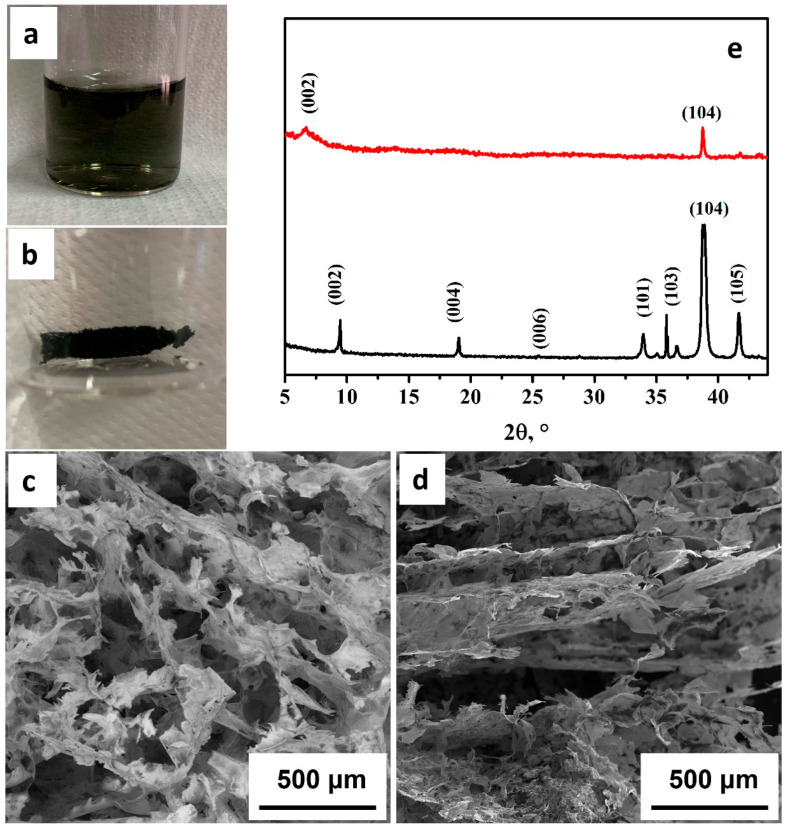
MXene morphology. (**a**) Dilution of the supernatant obtained in the MXene synthesis; (**b**) sponge obtained after the lyophilization process used in SEM analysis; (**c**) SEM image of the sponge observed crosswise; (**d**) SEM image of the longitudinal sample; (**e**) X-ray diffraction patterns of Ti_3_AlC_2_ (black) and Ti_3_C_2_T_x_ (red).

**Figure 2 membranes-12-01146-f002:**
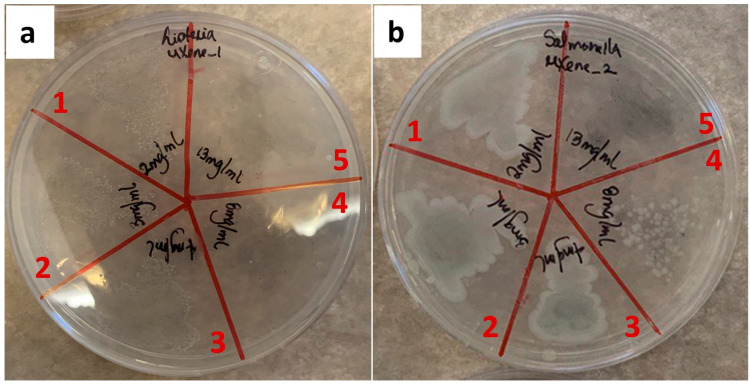
The minimum bactericidal concentration of MXene against (**a**) *L. monocytogenes*; (**b**) *S. enterica*. Red numbers correspond to the different tested concentrations as it follows: **1** corresponds to 2 mg/mL, **2** to 3 mg/mL, **3** to 4 mg/mL, **4** to 8 mg/mL and **5** to 13 mg/mL.

**Figure 3 membranes-12-01146-f003:**
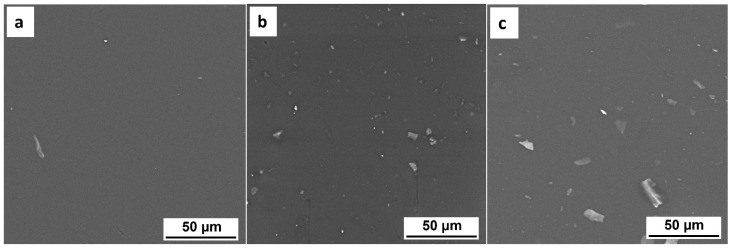
FESEM images of PLA/MXene composite films with (**a**) 0.3; (**b**) 1 and (**c**) 5 wt.% MXene.

**Figure 4 membranes-12-01146-f004:**
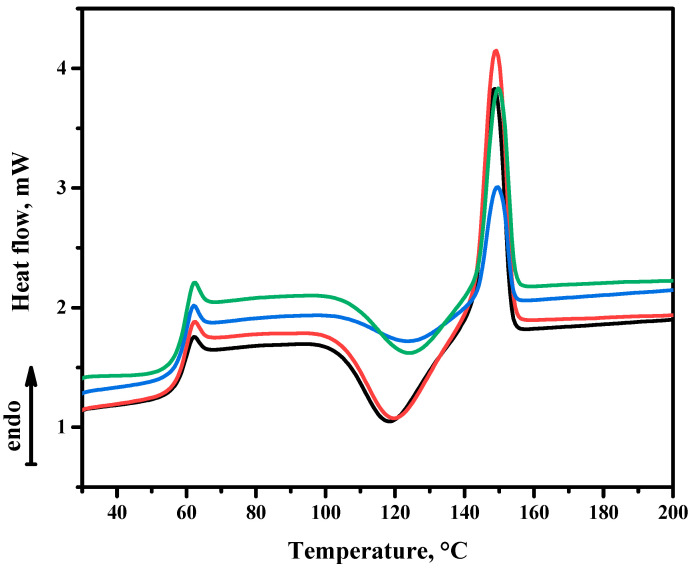
Differential scanning calorimetry of PLA and the obtained composites. PLA sample corresponds to the black line, PLA + 0.3 wt.% of MXene to the red line, PLA + 1 wt.% of MXene to the blue line and PLA + 5 wt.% to the green line.

**Figure 5 membranes-12-01146-f005:**
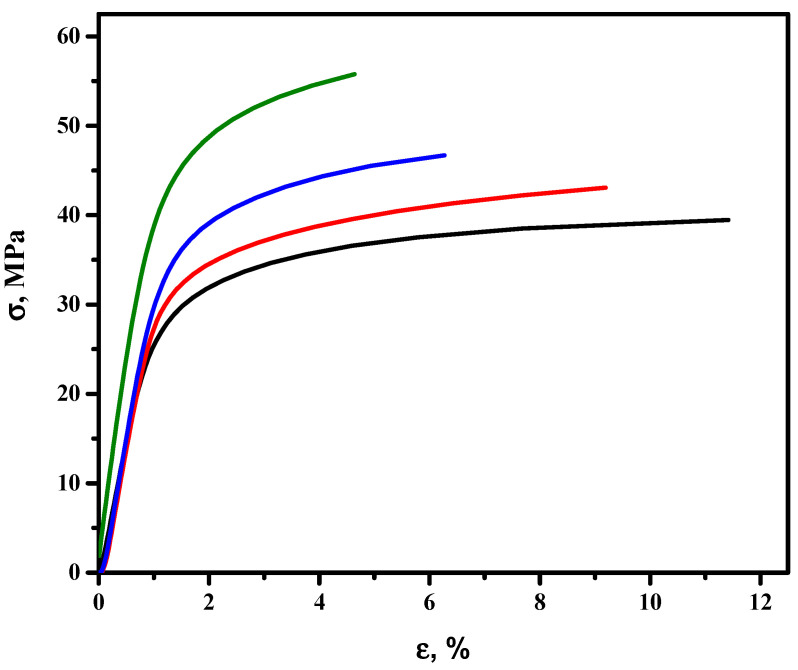
Stress vs. strain curves of the PLA and the composites with different loads (0.3, 1, 5 wt.%) of MXene particles. PLA sample corresponds to the black line, PLA + 0.3 wt.% of MXene to the red line, PLA + 1 wt.% of MXene to the blue line and PLA + 5 wt.% to the green line.

**Figure 6 membranes-12-01146-f006:**
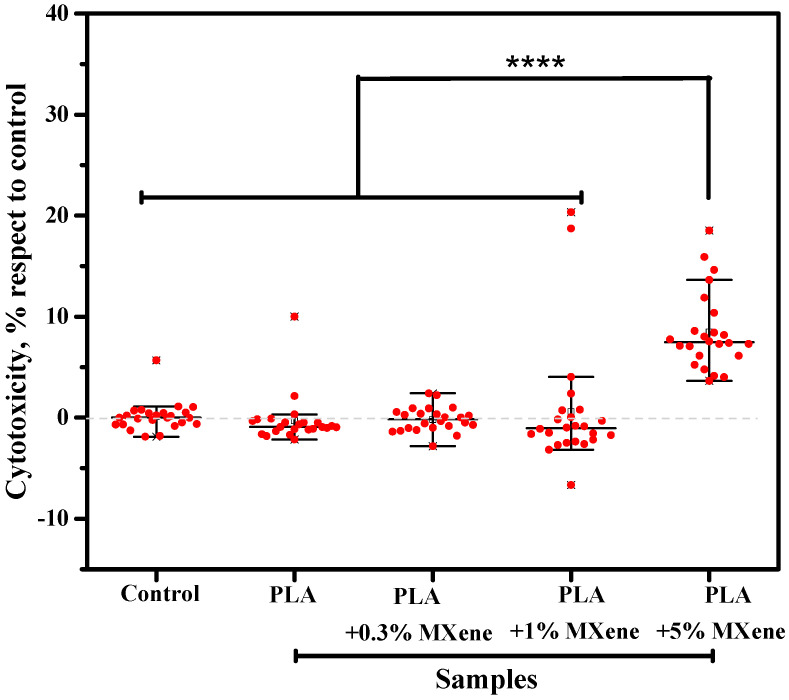
Cytotoxicity of the PLA and composite films with different loads (0.3, 1, 5 wt.%) of MXene particles.

**Figure 7 membranes-12-01146-f007:**
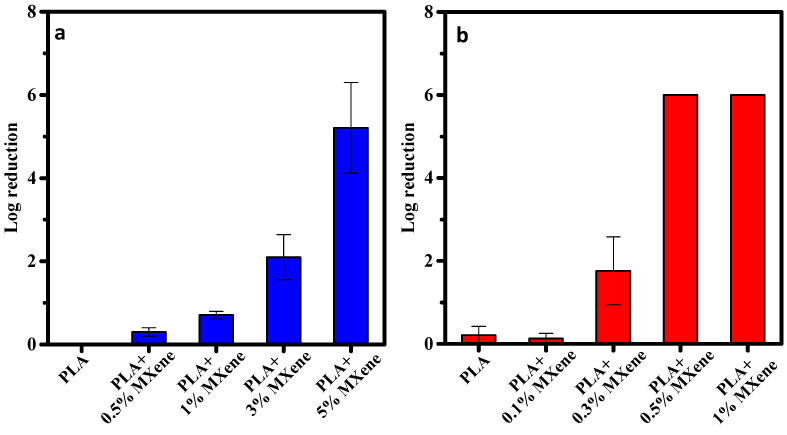
Logarithmic reduction for the PLA and the composite films with different concentrations of MXene (**a**) *Salmonella enterica* and, (**b**) *Listeria monocytogenes*.

**Table 1 membranes-12-01146-t001:** Results obtained from the calorimetric analysis of the PLA and composites.

Sample	Tg, °C	Tc_on set_, °C	Tm_on set_, °C	ΔHc, J/g	ΔHm, J/g
PLA	58	105	142	290	59
PLA + 0.3% MXene	58	104	142	301	67
PLA + 1% MXene	58	105	143	328	80
PLA + 5% MXene	58	105	143	350	81

**Table 2 membranes-12-01146-t002:** The decomposition temperature of PLA and nanocomposites.

Material	T_onset,_ °C	T_peak,_ °C
PLA	336	377
PLA + 0.3% MXene	327	366
PLA + 1% MXene	320	360
PLA + 5% MXene	320	358

**Table 3 membranes-12-01146-t003:** Elongation at break (ε_max_), tensile strength (σ_max_) and Young’s modulus (E) values of the PLA and the composites with different loads (0.3, 1, 5 wt.%) of MXene. The average thickness, width and length of the 5 replicas for each test sample are also represented.

Sample	Thickness, mm	Width, mm	Length, mm	σ_max_, MPa	ε_max_, %	E, GPa
PLA	0.05 ± 0.01	2.0 ± 0.1	8.78 ± 0.09	40 ± 10	14 ± 7	2.6 ± 0.4
PLA + 0.3% MXene	0.05 ± 0.01	2.13 ± 0.07	8.8 ± 0.1	41 ± 6	12 ± 3	2.9 ± 0.5
PLA + 1% MXene	0.05 ± 0.01	2.1 ± 0.09	8.80 ± 0.08	42 ± 8	10 ± 5	2.9 ± 0.5
PLA + 5% MXene	0.03 ± 0.01	2.04 ± 0.05	8.74 ± 0.01	68 ± 10	4.7 ± 0.2	4.8 ± 0.6

**Table 4 membranes-12-01146-t004:** Polymeric matrix/filler antibacterial systems reported in the literature having either PLA as the polymeric matrix or MXene as the filler.

Filler	Polymer	Bacteria	Log Reduction	Ref.
TiO_2_	PLA	*Escherichia coli* *Staphylococcus aureus*	~0.7 ~0.2	[32]
ZnO	PLA	*Escherichia coli* *Staphylococcus aureus*	~3.4 ~2.9	[33]
AgNPs	PLA	*Escherichia coli* *Listeria monocytogenes*	~10 ~9.8	[34]
Ag	PLA	*Staphylococcus aureus* *Escherichia coli* *Listeria monocytogenes* *Salmonella Typhimurium*	~2.7 ~7.2 ~6.1 ~7.4	[35]
AgNPs	PLA	*Escherichia coli* *Staphylococcus aureus*	~6.3 ~4.6	[36]
GO/ZnO	PLA	*Staphylococcus aureus* *Escherichia coli*	~2.1 ~1.6	[37]
ZnO NPs	PLA	*Escherichia coli* *Listeria monocytogenes*	~6.6 ~4.3	[38]
Ag, ZnO, TiO_2_	PLA	*Staphylococcus aureus* *Escherichia coli*	~0.4 ~8.0 (PLA-AgNPs)	[39]
NiO NPs supported on SiO_2_	PLA	*Listeria monocytogenes* *Salmonella*	~0.9 ~1.0	[40]
MXene	Nanocellulose	*Escherichia coli Staphylococcus aureus*	~1.6 ~1.8	[14]
MXene	PVDF	*Escherichia coli* *Bacillus subtilis*	~2.3 ~2.3	[51]
MXene	PLA	*Listeria monocytogenes* *Salmonella*	~6.0 ~5.2	This work

## Data Availability

Not applicable.

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
