# Peer review of "Antibacterial Capability of MXene (Ti3C2Tx) to Produce PLA Active Contact Surfaces for Food Packaging Applications"

_membranes, 2022, doi:10.3390/membranes12111146_

Round 1

Reviewer 1 Report

The manuscript presented by the Santos, X. et al is of interest, but I believe that there are still aspects that should be taken into consideration.

The introduction should be more extensive and include more bibliographic resources. Resources 1, 2 and 5 are repeated, and the citations do not specifically present results obtained in other studies.

The aim of the research is the development of materials for food packaging, but, according to the determinations and results, it is not clear how the material could be used for this purpose.

Moreover, there is no proof of obtaining an active and biodegradable packaging. Also, to obtain it, dichloromethane is needed as a solvent, a substance known to be irritating to the skin and eyes, even carcinogenic.

I don't think that the evaluation of cytotoxicity could eliminate this problem.

The obtained composition is not the most suitable, in this case. Maybe the material can be used for another purpose, but I don't think that the applicability in the food industry is a suitable one. The biodegradable character of the material is not highlighted. I think it would have been worth highlighting the anti-corrosive capacity of the material.

The part addressed to the discussions is brief and there are few reports in the specialized literature and other discoveries in the field.

Author Response

1- The introduction should be more extensive and include more bibliographic resources. Resources 1, 2 and 5 are repeated, and the citations do not specifically present results obtained in other studies.

R: thank you for your suggestion, references 1 and 2 constitute reviews that we consider that summarize the problems suffered by the food packaging industry, which lead us to the use of a sustainable material to obtain the active system that we propose. In the case of PLA, Table 2 (presented in results section) summarized several works in which similar PLA/reinforcement systems were explored for their potential application in food packaging industry. We decided to include these data in Table 2 of the results section in order to allow us to compare and highlight the achievements obtained.

In the case of reference 5, it is based on a study of listeriosis in Spain, which helps us to identify Listeria as one of the most important microorganisms to combat in the food industry.

Nonetheless, we have made changes in the introduction to improve the understanding of the raised problem. For example, references that demonstrate the use of biopolymers in the food packaging industry (cited in paragraph 2) have been added. Also, we added information corresponding to the biodegradation process of PLA (paragraph 2). Finally, we have added to the introduction the works based on PLA films with antibacterial properties (penultimate paragraph of the introduction).

2- The aim of the research is the development of materials for food packaging, but, according to the determinations and results, it is not clear how the material could be used for this purpose.

R: Thank you for your suggestion The primary focus is the study of the MXene effectiveness to be used as an inorganic filler that provides active properties to a polymeric matrix, in this case antibacterial properties. In this study, we have highlighted the antibacterial capability that the MXene provided on a PLA matrix, thus resulting in a antibacterial contact surface. Taking into account this and the possible mechanisms (found in literature) related to the antibacterial activity of MXene (cited in paragraph 8 of the introduction), we consider that the antibacterial property of these films is by direct surface contact, and thus will have to be placed in direct contact with food. Nevertheless, the application should not be only focused on the packaging itself, but also in food contact materials that cover multiple types of materials that may come in contact with food, and may become contaminated through manipulation (e.g., cutting board, cutting material, etc.).

3- Moreover, there is no proof of obtaining an active and biodegradable packaging. Also, to obtain it, dichloromethane is needed as a solvent, a substance known to be irritating to the skin and eyes, even carcinogenic.

R: thank you for your suggestion We consider that the films obtained in our research are "active materials" taking into account the definitions found in literature such as the one provided by Nikolic et. al that define an "active container" as that in which the particles added to the packaging material interact directly with the food and protect it from UV rays, oxygen or microbiological contamination [1]. Therefore, as MXene particles provide antibacterial properties to PLA, we consider that this an active system, can be used in food contact surfaces as an active material.

When we considered this study, we took into consideration the use of different polymeric matrices. However, it seemed especially important to us to propose an alternative in accordance with the current demands of the food industry for the substitution of plastics derived from petroleum by those that are friendlier to the environment. In this sense, PLA is one of those that has already been used precisely due to its biodegradability, a term used by different authors to characterize this polymer [2] [3] [4]. It is totally true that we do not highlight the biodegradability of the material, precisely because so far we do not have our own degradation results for our composites. However, taking into account the suggestions proposed to our manuscript, we have added in paragraph 2 of the introduction information corresponding to the PLA biodegradation process. It should be noted that biodegradation studies have already been carried out under composting conditions for the PLA/TiO2 system, in which the authors reached the conclusion that the addition of the particles to the polymeric matrix causes an increase in the rate of degradation of the PLA [4].

We understand the concern of using the solution casting method (with dichloromethane as solvent) to obtain the composite films. We do not consider the use of this method to replace those currently used to obtain films in the food packaging industry, for which dichloromethane cannot be used. We wanted to study the effectiveness of MXenes as an active inorganic filler and for this solution casting and the PLA polymer matrix were selected due to the simplicity in making the PLA/MXene films. This reason added to the fact that the thermal properties do not show any other degradation process corresponding to the added nanoparticles and that the mechanical properties are not profoundly affected, allow us to propose the addition of these particles as an additive to PLA, a polymer already used in the industry of food, to confer the active role of the MXene.

Therefore, we chose the solution casting method to obtain the composites because it is one of the most used for the design and study of systems with potential applications in food packaging technology due to its simplicity [5] [6] [7] [8]. Even systems based on PLA with dichloromethane as solvent have been used for this purpose [9] [10] [11].

We used dichloromethane, because between all the solvents found in the literature for solution casting of PLA, it was the easiest to remove, evaporating more than 90% in the first 90 minutes at room temperature [12]. Even in our case, the films were dried for 5 days in an oven at 40 ºC to eliminate any influence of the solvent on the analysed properties. In the case of the presence of traces of the solvent in the obtained composites, they did not influence either the thermal or mechanical properties (since no significant changes are observed in the Tg or in the mechanical parameters obtained compared with those reported in the literature for PLA). We again emphasize that we are not proposing the use of dichloromethane in the food industry, but we are proposing the addition of MXene particles as an interesting additive that would give PLA, antibacterial properties. Indeed, this study gave us the necessary information to begin the next step of adding the MXene by melt extrusion process into the PLA.

Además, en la técnica EDS (ver figura abajo), aplicada a compuestos, no se observa la presencia de cloro en las áreas correspondientes al polímero que podría derivarse de la presencia de diclorometano (como se muestra en la imagen y la tabla de ejemplo). De esta manera, consideramos que las propiedades antibacterianas encontradas en nuestros materiales no se deben a la presencia de diclorometano en las muestras (considerando también las muestras de control de PLA), sino a las propiedades biocidas que proporcionó el MXene.

Figura 1. Análisis EDS en la película compuesta de PLA.

4- No creo que la evaluación de la citotoxicidad pueda eliminar este problema.

R: gracias por su sugerencia La citotoxicidad de los materiales no se estudió para evaluar la influencia del diclorometano en las muestras. Como explicamos en el punto anterior, consideramos que si hay trazas de la misma en los composites, estas no son las que determinan las propiedades antibacterianas y citotóxicas.

El concepto de realizar ensayos de citotoxicidad en nuestras muestras se basa en la posible aplicación de MXene como aditivo al PLA para la producción de un material activo en la industria del envasado de alimentos y/u otros materiales en contacto con alimentos. En este estudio podemos observar que incluso a las concentraciones más altas estudiadas de MXene (5 en peso), los materiales no son citotóxicos.

5- La composición obtenida no es la más adecuada, en este caso. Tal vez el material se pueda utilizar para otro propósito, pero no creo que la aplicabilidad en la industria alimentaria sea adecuada. No se resalta el carácter biodegradable del material. Creo que hubiera merecido la pena destacar la capacidad anticorrosiva del material.

R: gracias por su sugerencia, como dijimos en los puntos 1 y 3, elegimos PLA como matriz polimérica debido a su biodegradabilidad ya conocida y estudiada. Este tema no se ha tratado en profundidad en la introducción porque hasta ahora no hemos realizado ensayos de biodegradabilidad sobre los composites obtenidos. Debido a esto, en el párrafo 2 del manuscrito hemos agregado información encontrada en la literatura sobre cómo ocurre la biodegradación del PLA.

En cuanto a la aplicación dentro de la industria del envasado alimentario, volvemos a destacar que existen múltiples formas de aplicación. Otros son, por ejemplo, su uso en otros materiales/superficies en contacto con alimentos (no necesariamente un envase de alimentos), como tablas de cortar, etc. Queríamos estudiar el papel activo de la adición de partículas de MXene en una matriz polimérica. Para ello hemos conseguido obtener propiedades antibacterianas como su papel activo dentro de una matriz de PLA. En cuanto al sistema PLA/MXene en sí, no estamos proponiendo la aplicación de este sistema compuesto como producto final, sino que se utilizó el PLA como matriz elegida para evaluar la efectividad del MXene.

Las propiedades anticorrosivas tampoco han sido evaluadas porque proponemos un material con propiedades antibacterianas que estaría en contacto con los alimentos, y no como recubrimiento para metales susceptibles a la corrosión.

6- The part addressed to the discussions is brief and there are few reports in the specialized literature and other discoveries in the field.

R: thank you for your suggestion, additional discussions were implemented in the section 3.2.5 with respect to the antibacterial properties of the developed composites and changes in the introduction were made. In addition, the session corresponding to the discussion of the results of the mechanical properties (3.2.2) was modified as suggested by reviewer 3 (point 16).

Esperamos que con los cambios realizados en el manuscrito como consecuencia de las sugerencias propuestas a nuestro trabajo, los resultados obtenidos en nuestra investigación hayan sido explicados, incluso verificados con otras fuentes con el fin de analizar las tendencias o señales obtenidas en cada una de las técnicas utilizadas en la caracterización tanto de las partículas como de los compuestos.

Reviewer 2 Report

Authors present a simple method to prepare food packaging from PLA and MXenes. These materials meet the basic requirements for packaging and provide microorganism resistance. However, there are two major issues with this work. Firstly, MXene are are quite expensive, making the argument for their use in food packaging difficult. The argument that they provide resistant to bacteria such as listeria is not enough to justify their cost as cheaper alternatives, such as silver or copper, also perform quite well. Additionally, MXene are not biodegradable, so difficult to claim that is an attribute of these materials and again justify the usage of Mxene. PLA itself is not actually biodegradable, only compostable, as it does not break down naturally in a standard environment. The biodegradability of the material would need to be tested in order for this claim to remain in the manuscript. Please address both these concerns.

Introduction, page 2: Update the paragraph on smart packaging. The definition is used quite broadly in the context of this work, as the addition of a antimicrobial agent does not necessarily constitute a “smart material”.

Why was a freeze drying methodology chosen for drying? This method is expensive and not necessarily viable for a large scale production.

The methodology used to create the films is incorrect for food packaging. Materials would not be solution cast in organic solvent and doctor bladed onto surface. How can the authors argue use in this application?

Concentration is MXenes in the final composites is quite high for activity, especially considering the toxicity and antimicrobial performance. Authors should have explored more lower amounts.

Figure 4, please provide a label on the actual graph.

Not all materials presented in Table 1 are MXenes, update the title.

Author Response

1- Authors present a simple method to prepare food packaging from PLA and MXenes. These materials meet the basic requirements for packaging and provide microorganism resistance. However, there are two major issues with this work. Firstly, MXene are are quite expensive, making the argument for their use in food packaging difficult. The argument that they provide resistant to bacteria such as listeria is not enough to justify their cost as cheaper alternatives, such as silver or copper, also perform quite well. Additionally, MXene are not biodegradable, so difficult to claim that is an attribute of these materials and again justify the usage of Mxene. PLA itself is not actually biodegradable, only compostable, as it does not break down naturally in a standard environment. The biodegradability of the material would need to be tested in order for this claim to remain in the manuscript. Please address both these concerns.

R: thank you for your suggestion , it is totally true that PLA wastes could only be treated under composting conditions (accelerated biodegradation processes) because under standard conditions the biodegradability of this polymer could take many years. However, in all the consulted sources, including those in which the biodegradability of PLA is studied under composting conditions, they start from the premise that the polymer is a biodegradable material. We consider the clarification that has been proposed to us totally timely, so we have decided to change the title of the article to "Antibacterial capability of MXene (Ti3C2Tx) to produce active contact surfaces".

This one is not a property that we study directly in our work, but rather we speculate that our composites remaing biodegradable due to the fact that the matrix consists of a biodegradable polymer. However, as we stated in the point 3, we added information related to the biodegradability of PLA in the introduction. In addition, we consider that the study of this property in our composites is a good starting point for further work to assess whether if due to the well-known hydrophilicity of MXene, it positively influences the biodegradability of PLA, as has been observed in the system. PLA/TiO2 cited above [4].

With respect to the MXene, we consider that one of its main attributes is the biocompatibility that the Ti3C2Tz-type MXene has shown to date. For example, Chen et. al have studied membranes based on PLA and MXene (Ti3C2Tz), obtained by solution casting (using dichloromethane as solvent) with potential biomedical applications in guided bone regeneration. Where the authors concluded that the membranes loaded with MXene improve the proliferation of preosteoblasts and cell viability with respect to those of pristine PLA. That is why in the manuscript we have added information related to the choice of this nanofiller that we consider justifies the use of MXene as a possible candidate for obtaining active materials with potential applications in the food packaging industry.

As we know, the “MXene (Mn+1XnTz) are a relatively new family of two-dimensional inorganic materials made up of carbides or nitrides obtained from the selective exfoliation of the “A” layers of the MAX phases. These materials have a high specific surface area, good electrical conductivity, hydrophilicity, and antibacterial activity, which has promoted their use in a wide range of applications, both from the electronic and sensor point of view, as well as for biomedical applications. Therefore, they have become one of the most versatile materials during the last decade, whose synthesis, properties and the exploration of new applications have become one of the areas of greatest interest in science and technology. Since up to now we have not found specialized literature that relates MXene as a possible additive for obtaining active materials in the food packaging industry, it has seemed to us an important and interesting topic to study.

Although it is true that due to the novelty of these materials, their cost is still high, we believe that due to their excellent and versatile properties and the interest that exists in science and industry in their insertion, this problem can be resolved. Furthermore, the provided activity was obtained at very low loading contents (in the case of Listeria). Nonetheless, we are aware of the existence of works in the literature such as that of Shuck et. al, in which they propose a methodology for the scalable synthesis of MXene Ti3C2Tx [13].

2- Introduction, page 2: Update the paragraph on smart packaging. The definition is used quite broadly in the context of this work, as the addition of a antimicrobial agent does not necessarily constitute a “smart material”.

R: thank you for your suggestion , we completely agree with this suggestion, which is why we have proceeded to eliminate the term “smart” to “active” in the paragraph in question.

3- Why was a freeze drying methodology chosen for drying? This method is expensive and not necessarily viable for a large scale production.

R: thank you for your suggestion, as we have commented in the manuscript in the results and discussion section (section 3.1) “This drying process avoids the stacking of the MXene nanosheets together, which would hinder future dispersion processes, and avoids the oxidation of MXene in aqueous dispersions.” This makes it easier for us to obtain nanoparticles in the laboratory.

For example, in the work cited in point 7, which is based on the scalable synthesis of MXene Ti3C2Tx to obtain nanoparticles, they use vacuum drying [13]. In the following figure we can observe the system that they propose for the synthesis.

Figure 2. Schematic of MXene synthesis

4- The methodology used to create the films is incorrect for food packaging. Materials would not be solution cast in organic solvent and doctor bladed onto surface. How can the authors argue use in this application?

R: thank you for your suggestion, please refer to the answer to point 3 of first reviewer.

The primary focus is this study was to evaluate the MXene effectiveness to be used as an inorganic filler that provides active properties to a polymeric matrix, which in turn could be used as a food contact material. In this study, we have highlighted the antibacterial capability that the MXene provided on a PLA as the chosen matrix. We again emphasize that we are not proposing the use of dichloromethane and the process of solution casting to develop films to be applied in food packaging (as we known that they are not commonly used), but we indeed used them in order to obtain films under rapid and simple laboratory conditions to test if MXenes provided any active role to the PLA. Finally, we are proposing the addition of MXene particles as an interesting additive that would give antibacterial properties to PLA. Indeed, this study gave us the necessary information to begin the next step of adding the MXene by melt extrusion process into the PLA.

As for the application within the food packaging industry, we again highlight that there are multiple ways of application. Others are, for example, their use in other food contact materials/surfaces (not necessarily a food packaging), such as cutting boards, etc.

5- Concentration is MXenes in the final composites is quite high for activity, especially considering the toxicity and antimicrobial performance. Authors should have explored more lower amounts.

R: thank you for your suggestion, antibacterial tests were carried out to determine at what concentration of MXene the complete elimination of the studied bacteria was achieved. As shown by the data concentrations of 0.5% allowed the total elimination of Listeria and at least 3% is necessary for the total elimination of Salmonella. Considering the cytotoxicity results, a concentration of 1% can be represented as the limit in which the toxicity starts to increase. Giving this data concentrations as low as 0.5% are sufficient to give antibacterial active, however more effective against Listeria. Nontheless and bearing in mind that in later works we want to study these properties with food products prone to presenting this type of microorganisms, eventually these quantities will be optimized.

6- Figure 4, please provide a label on the actual graph.

R: thank you for your suggestion, both in the graphs presented in the manuscript and in the supplementary material, the legend of each of the lines is described in the caption of the image.

“PLA sample corresponds to the black line, PLA + 0.3 wt.% of MXene to the red line, PLA + 1 wt.% of MXene to the blue line and PLA + 5 wt.% to the green line.”

7- Not all materials presented in Table 1 are MXenes, update the title.

R: thank you for your suggestion, the name of Table 2 (previously Table 1) has been changed to:

“Polymeric matrix/filler antibacterial systems reported in the literature having either PLA as the polymeric matrix or MXene as the filler”

Reviewer 3 Report

The manuscript describes a method of using MXene with PCL for food packaging and reported the antimicrobial effects. Overall, the manuscript is well presented. I have the below suggestions for further improvement.

1. How the biodegradability is going to be affected by this method? Would be nice if the authors could provide some information in the Introduction based on previous findings. 

2. Though the authors have referred to supplementary figures and tables in the manuscript (such as S1, T2), I was not able to see the file in the system.

3. It is also better to include the values of tensile strength, elongation and modulus in the main draft, rather than in the supplementary, though the representative stress-strain curve can be placed in the supplementary.

4. What sort f food packaging is suggested? What is the next step forward? These can be included in the Conclusions.

Author Response

  1. 1. How the biodegradability is going to be affected by this method? Would be nice if the authors could provide some information in the Introduction based on previous findings. 

R: thank you for your suggestion, information related to PLA biodegradability has already been added to the manuscript, and the role of what the incorporation of additives does to the PLA biodegradability. In addition, please refer to thepoints 3 and 7 that explain this issue.

  1. 2. Though the authors have referred to supplementary figures and tables in the manuscript (such as S1, T2), I was not able to see the file in the system.

R: thank you for your suggestion, we are not aware of why the supplemental material file cannot be accessed.

  1. 3. It is also better to include the values of tensile strength, elongation and modulus in the main draft, rather than in the supplementary, though the representative stress-strain curve can be placed in the supplementary.

R: thank you for your suggestion, the table corresponding to the mechanical properties of the films has been inserted in the manuscript (formerly Table S3 now Table 1 of the manuscript).

  1. 4. What sort f food packaging is suggested? What is the next step forward? These can be included in the Conclusions.

R: thank you for your suggestion, these issues have been addressed in points 2, 3, 7, 11. Notwithstanding, some conclusions from what was previously discussed have been added to the conclusions section of the manuscript.

Round 2

Reviewer 2 Report

Revisions are acceptable, thank you